# Use of Alkali-Activated Slag as an Environment-Friendly Agent for High-Performance Stabilized Soil

**DOI:** 10.3390/ma16134803

**Published:** 2023-07-03

**Authors:** Qinghua Huang, Guohui Yang, Chenzhi Li, Mingzhi Guo, Tao Wang, Linhua Jiang

**Affiliations:** 1College of Mechanics and Materials, Hohai University, Nanjing 210098, China; 221308020003@hhu.edu.cn (Q.H.); ghyang035@hhu.edu.cn (G.Y.); mz.guo.mz@connect.polyu.hk (M.G.); 15850651297@163.com (T.W.); 2Department of Structural Engineering, College of Civil Engineering, Tongji University, Shanghai 200092, China; chenzhili@tongji.edu.cn

**Keywords:** alkali-activated slag, soil-stabilized agent, waste utilization, mechanical properties, durability

## Abstract

Discharged slag not only occupies a large amount of land for disposal, but also causes serious environmental pollution. The use of alkali-activated slag (AAS) instead of cement as a soil-stabilization agent is beneficial for industrial waste disposal and energy conservation, which complies with the concept of green and low-carbon sustainable development in the construction industry. In this study, the compressive strength, water permeability coefficient, chloride migration coefficient and sulfate resistance of alkali-activated slag-stabilized soil (AASS) were evaluated, and compared with those of cement-stabilized soil (CSS). The hydrated crystalline phases and microscopic pore structures were analyzed by X-ray diffraction, electrochemical impedance spectroscopy (EIS) and mercury intrusion porosimetry (MIP) tests, respectively. The results indicate that, compared with CSS, AASS exhibits a higher compressive strength, lower water permeability, chloride migration coefficient and better resistance to sulfate attack, with the optimum dosage higher than 10 wt.%. The results of the MIP analysis show that the addition of AAS reduces the porosity by 6.47%. The combined use of soil and AAS proves to be a viable and sustainable method of waste utilization and carbon emission reduction in the construction industry, which provides a practical path towards carbon peaking and carbon neutrality.

## 1. Introduction

The rapid development of infrastructure construction creates an ever-increasing demand for river sand, gravel stone, cement and other engineering materials, resulting in the depletion of natural resources and causing serious threats to the eco-system [1,2,3]. To alleviate the environmental burden and realize the sustainable development of society, the utilization of resource-saving and environment-friendly engineering materials is required [4,5,6]. In recent years, soil resources have been recycled and utilized in coastal and offshore areas of China, since the coastline is abundant in silt soil resources [7]. After modification by a stabilization agent, the silt soil exhibits an increased compactness and decreased water content, along with an improved physical property and mechanical performance [8]. Previous studies demonstrated that the stabilized soil can be used as a preferable construction material due to its good volume stability, mechanical properties, freeze-thaw resistance and low engineering cost [9,10,11,12,13].

In long-term engineering practice, cement, lime and fly ash are often used to solidify soil [14,15]. However, the soil stabilized by these agents suffers from low early-age strength, large dry shrinkage and weak water resistance, which seriously compromises its durability performance and engineering safety, especially for the use of stabilized soil in underground engineering. The modification effect of these stabilization agents is closely related to the soil property. For clay, saline soil, organic soil and sludge with high plasticity index, the stabilization effect is significantly limited [16,17]. Furthermore, the production of cement and lime consumes a huge amount of energy and has a colossal carbon footprint, which is not in line with energy conservation, emission reduction and environmental protection [18,19,20]. Thus, an effective and eco-friendly stabilization agent for soil engineering is required.

Blast-furnace slag is a by-product derived from the production of smelting pig iron. On average, the production of 1 t smelting pig iron leads to the generation of 0.3–0.9 t discharged blast-furnace slag [21,22,23]. If not utilized in an environmentally friendly manner, the discharged slag will not only occupy a large amount of land for disposal, but also cause serious ground water pollution [24]. The ensuing environmental and economic issues will bring serious threats to the eco-system and sustainable development of our society. Previous studies have shown that the use of slag as a soil stabilization agent can achieve a curing effect comparable to that of cement [25,26]. Also, in cement-based materials, remarkable improvements in mechanical properties and durability can be achieved if the potential activity of slag can be activated by some alkaline substances, i.e., alkali-activated slag (AAS). Compared with ordinary Portland cement, AAS possesses a higher strength, shorter setting time, better resistance to chemical erosion and lower hydration heat owing to its potential hydraulic activity [27,28,29,30,31,32]. Therefore, it can be inferred that AAS has great potential to be used as a stabilization agent for soil, which may provide a reasonable method for the utilization of the industrial waste slag and development of resource-saving and environment-friendly manufacturing industries. However, few investigations on the durability of AAS-stabilized soils, such as their resistance to chloride and sulfate ion attack, have been reported in this field.

Some scholars have studied the properties of alkali-activated slag and fly ash [33,34]. Reza et al. [35] studied the effects of NaCl on the microstructure and mechanical properties of stabilized sandy soil with alkali-lithium-activated ash slag and confirmed that the alkali-lithium-activated slag could replace Portland cement for soil stabilization engineering in saline-alkali environments. Hania et al. [36] investigated the potential of alkali-active powdered blast furnace slag and volcanic ash as green binders in clay stabilization projects. The total binder content was fixed at 20% of the soil mass. When the content of volcanic ash reached 70% of the total binder, the strength was improved, and the F-T and W-D durability tests were carried out to study the properties of the stabilized soil. According to the aforementioned above, the properties on the mechanical strength, and F-T and W-D durability tests have been investigated in previous studies, while other durability studies, such as water permeability coefficients, chloride migration coefficients and sulfate resistance of alkali-activated slag and fly ash in the process of stabilizing the soil were lacking, which is of great significance to its application in engineering.

In this study, AAS was used as a stabilization agent for soil at different dosages (5 to 20 wt.%). The compressive strengths, water permeability coefficients, chloride migration coefficients and sulfate resistance of alkali-activated slag-stabilized soil (AASS) and cement-stabilized soil (CSS) were evaluated and compared. Mercury intrusion porosimetry (MIP), electrochemical impedance spectroscopy (EIS) and X-ray diffraction (XRD) analyses were conducted to examine the pore structure and crystalline phase composition of AASS in order to decipher the stabilization mechanisms from a microscopic perspective. The findings from the present study will not only provide new insights into the reuse of industrial slag wastes, but also facilitate the wide application of soil for cleaner production purposes.

## 2. Experimental Program

### 2.1. Materials

P. C 32.5 Portland cement, in accordance with Chinese standard GB 175-2007 [37], and backfill soil were employed as raw materials, and their basic properties are provided in Table 1 and Table 2. The chemical composition of the blast furnace slag is shown in Table 3. Sodium silicate with a modulus of 3.2 was used in the test, and its chemical composition is summarized in Table 4.

The blast furnace slag was activated by sodium silicate and NaOH solution. The sodium silicate modulus was adjusted to 1.5 (Na_2_O·1.5SiO_2_) by adding NaOH solution. According to a preliminary study [38], the alkali equivalent was set at 7% to achieve an optimum activation effect. All the soil samples were oven dried and ground to meet the requirement of testing. The curing agent and soil were stirred in the mixer for 1 min, and then gradually mixed evenly with water (30% of the weight of soil), added during the stirring process. The stirring was continued for 2 min, and then they were cast into shape, the surfaces were covered with plastic wrap, they were removed from the mold for 24 h, and were then the surfaces were again covered with plastic wrap for maintenance until aged. Alkali-activated slag was added at dosages of 5 wt.% to 20 wt.% (by mass of soil), and samples with the same dosage of cement addition were prepared for comparison, as shown in Table 5.

### 2.2. Test Procedures

#### 2.2.1. Compressive Strength

Cube test blocks with dimensions of 70.7 mm × 70.7 mm × 70.7 mm were employed for the compressive strength test using a hydraulic universal testing machine. Due to the relative low strength of stabilized soil, the loading speed was maintained at a low range of 0.05 to 0.15 MPa/s. The compressive strength was measured after the soil specimens were standard cured for 3, 7, 28 and 90 d. Three duplicates were prepared for each mixture group to obtain reliable statistical results.

#### 2.2.2. Water Permeability Coefficient

Circular truncated cone specimens with a height of 30 mm, upper diameter of 70 mm and lower diameter of 80 mm were prepared for the test. After cured to the target age, the specimens were oven dried for 12 h and then cooled to ambient temperature. The side surface of the specimen was sealed with a layer of silicone rubber before being placed in the water penetrameter. The initial water pressure was set at 0.2 MPa and kept for 2 h, and then the water pressure was increased by 0.1 MPa every hour until 1.5 MPa. Subsequently, the specimens were spilt into two halves, and the water penetration depth was measured. The water permeability coefficient (*K_w_*) can be obtained using Equation (1):(1)$$K_{w}=\frac{\alpha {D_{m}}^{2}}{2TH}$$
where *D_m_* denotes the average water penetration depth (cm), *H* represents the water column height (cm), *T* is the loading duration, taken as 1 h, and *α* stands for the water uptake factor of stabilized soil, usually taken as 0.03.

#### 2.2.3. Chloride Migration Coefficient

Chloride migration coefficients were determined using rapid chloride migration (RCM) test in compliance with NT BUILD 443 (1995). Cylindrical specimens with dimensions of Φ 50 mm × 100 mm were employed for the test. The specimens were covered with plastic film immediately after molding and moved to a standard curing room. After 24 h, the specimens were demolded and immersed in a pool for curing. The DC power supply was adjusted to 30 ± 0.2 V during the test. The rubber cylinder containing the specimens was placed into the test tank, and an anode plate was installed. Then, about 300 mL of 0.2 mol/L KOH solution was injected into the rubber cylinder to immerse the surface of the anode plate and the specimen. A 0.2 mol/L KOH solution containing 5 wt.% NaCl was injected into the test tank until it was flush with the liquid level of the KOH solution in the rubber cylinder. The chloride penetration depth was obtained by measuring the average depth of the white precipitates on the fractured surface after the specimens were split into two halves and sprayed with 0.1 mol/L AgNO_3_ solution. Details regarding the determination of chloride migration coefficient can be found in our previous work [2].

#### 2.2.4. Sulfate Resistance

Cube soil specimens with dimensions of 70.7 mm × 70.7 mm × 70.7 mm were employed for the test after being standard cured for 90 d. Prior to the sulfate exposure, the compressive strengths of the specimens were determined as reference values. Subsequently, the specimens were soaked in 5 wt.% Na_2_SO_4_ solution for 14, 28 and 60 d. During the immersion period, the pH value of the solution was maintained between 6 and 8, and the temperature was kept at 25–30 °C. The sulfate resistance coefficient (*K_s_*) was defined using the following equation:(2)$$K_{s}=\frac{f_{cn}}{f_{c90}}$$
where *f_c_*_90_ refers to the 90 d compressive strength of soil specimen (MPa) and *f_cn_* denotes the compressive strength of specimen after being exposed in Na_2_SO_4_ solution for n days (MPa).

#### 2.2.5. Test

The EIS was tested by the Single Sine standard template in the PowerSine module of PARSTAT 2273 electrochemical workstation. The scanning frequency of the EIS test was 100 kHz~10 mHz. Sinusoidal AC voltage with amplitude of 5 mV was used for impedance test signal, and 50 points were selected during scanning. MIP test was performed to evaluate the porosity and pore size distribution of AASS and CSS samples using an AutoPore IV 9500 automatic mercury injection Instrument. The maximum mercury pressure was 50,200 PSI (Pounds per Square Inch) and the determined pore diameter ranged from 202 μm to 4.2 nm. The mercury contact angle and surface tension were 140° and 0.485 N/m, respectively. The crystalline phase compositions of AASS and CSS samples were measured by a Japanese Science D/Maxr B-type X-ray diffractometer (Cu kα, 40 kV voltage, 100 mA current) The diffraction angles were scanned from 5° to 65° (2θ).

## 3. Results and Discussion

### 3.1. Effect of AAS Addition on Compressive Strength of Soil

#### 3.1.1. Effect of AAS Dosage

Figure 1 shows the compressive strength of AASS and CSS samples at 3, 7, 28 and 90 d. In general, the compressive strength of AASS was lower than that of CSS at 3 and 7 d. This is because the hydration process of cement begins upon its contact with water, while the alkali-activated reaction only begins in a high alkalinity environment. A higher water content resulted in a decreased pH value of the alkali solution of sodium silicate, thereby delaying the strength development of the slag.

With increasing curing age, the slag came into contact with more sodium silicate. Owing to the alkali-activated action, the silicate and aluminate structures in the slag glass were destroyed by OH^−^, which facilitated the hydration and led to a remarkable increase in later-age strength, especially for samples containing 15 and 20 wt.% of AAS. Compared with CSS, AASS with 5 and 10 wt.% had a lower later-age strength, indicating that a low dosage of AAS was not enough to completely activate the latent hydraulic activity of slag. When the AAS content was increased to 15 and 20 wt.%, the corresponding 28 d strength was increased to 4.45 MPa and 6.06 MPa, and the 90 d strength was increased to 7.16 MPa and 10.62 Mpa, respectively, which was obviously higher than those of the CSS samples with the same dosages of cement. It can be seen that the effect of AAS dosage on the strength development of stabilized soil became more remarkable with increasing dosages.

#### 3.1.2. Effect of Curing Age

Figure 2 gives a comparison of the strength development in the AASS and CSS samples at different curing ages. Compared with CSS samples, AASS samples exhibited a slow strength increase at 3 and 7 d, while a notable increase in 28 d and 90 d strength could be observed. This may be due to the rapid hydration process of cement in the early stage, while the Si-O and Al-O bonds of slag can only be destroyed under the activation of OH^−^. After breaking, the vitreous body structure was destroyed, leading to the polymerization of [SiO4]^4−^. At this time, the monomer content of [SiO4]^4−^ decreased, and the polymer content increased, consequently resulting in the formation of C-S-H or C-A-H gels.

### 3.2. Effect of AAS Addition on Water Permeability Coefficient of Soil

#### 3.2.1. Effect of AAS Dosage

Variations of water permeability coefficients of samples with different dosages of AAS and cement are revealed in Figure 3. The water permeability coefficients gradually decreased with increasing AAS and cement contents, indicating that both the two stabilization agents were effective in enhancing the water resistance of the soil. The water permeability coefficients of AASS were higher than those of CSS at dosages of 5 wt.% and 10 wt.%, and lower than those of CSS at dosages of 15 wt.% and 20 wt.% regardless of the curing ages. AAS glass mainly consists of calcium-rich and silicon-rich phases, and the former is the structure-forming body which maintains the structural stability of AAS glass. When the AAS comes into contact with water glass, the calcium-rich phases on the slag surface transform to Ca(OH)_2_ and Mg(OH)_2_ under the activation of OH^−^ in sodium silicate solution, thereby destroying the glass structure and fracturing the Si-O-Si, Si-O-Al and Al-O-Al bonds. The generated ions as a result of the disintegration of the glass structure enter the soil solution and recombine with the active particles, such as ion exchange and granulation, consequently leading to the formation of new hydration products. With increasing AAS content, more slag participates in the hydration reaction and generates more high-strength hydration products, which bind the soil particles together and fill the interstices inside the matrix. As a result, a more compact AAS-soil skeleton structure forms, contributing to enhanced water resistance.

#### 3.2.2. Effect of Curing Age

Figure 4 presents a comparison of the water permeability coefficients in AASS and CSS at different curing ages. The water permeability coefficients of AASS and CSS gradually decreased with increasing curing ages, suggesting a denser microstructure owing to the proceeding of hydration, which generates more hydration productions that can fill the large capillary pores inside the soil. A remarkable reduction in permeability coefficients could be found from 28 to 60 d, but the reduction became insignificant from 60 to 90 d, suggesting that the hydration process became stable.

### 3.3. Effect of AAS Addition on Chloride Migration Coefficient of Soil

#### 3.3.1. Effect of AAS Dosage

Figure 5 reveals that both cement and AAS were effective in decreasing the chloride migration coefficients. At dosages of 5 and 10 wt.%, there was no discernible difference between the chloride migration coefficients of CSS and AASS. For AASS samples, a more remarkable effect on chloride resistance was achieved at an AAS content of above 10 wt.%, reflected by a sharp decrease in migration coefficients. When the AAS content was increased from 10 wt.% to 15 wt.%, the 28 d, 60 d and 90 d migration coefficients were decreased by 37.3%, 45.7% and 50.3%, respectively (much higher than those of the CSS samples). This suggested that the solidifying effect of ASS became more significant with increasing dosages, leading to the densification of the soil matrix and thereby enhancing the chloride resistance.

#### 3.3.2. Effect of Curing Age

Figure 6 presents a comparison of the chloride migration coefficients in AASS and CSS at different curing ages. In general, the chloride migration coefficients decreased with increasing curing ages, which was consistent with the variation trend of water permeability coefficients. With increasing curing ages, the difference between the migration coefficients of AASS and CSS samples became more notable, especially at 90 d. This can be attributed to the latent hydraulics of AAS, which promoted the later-age hydration and resulted in a more compact soil matrix with lower porosity.

### 3.4. Effect of AAS Addition on Sulfate Resistance of Soil

#### 3.4.1. Effect of AAS Dosage

The sulfate resistance coefficient was defined according to the variation in compressive strength before and after sulfate exposure. As shown in Figure 7, sulfate exposure significantly undermined the compressive strength of CSS samples, but enhanced the compressive strength of AASS samples. In CSS samples, cement hydration led to the formation of calcium hydroxide. The intruded sulfate ions reacted with calcium hydroxide to form a large amount of dihydrate gypsum and ettringite, which significantly increased the solid phase volume and caused microcracks owing to the expansion stress, thereby compromising the mechanical properties [39,40]. In contrast, the hydration of AAS produced a lower amount of calcium hydroxide due to the pozzolanic reaction, and the main hydration products of AAS are C-A-S-H, which exhibited a lower Ca-Si ratio and denser structure, thereby effectively limiting the volume expansion [41]. With increasing AAS dosages, the compressive strength slightly increased, indicating better sulfate resistance.

#### 3.4.2. Effect of Exposure Time

Figure 8 gives a comparison of the sulfate resistance coefficients in AASS and CSS at different exposure times. The sulfate resistance coefficients of AASS increased with increasing exposure times. For example, the addition of 20 wt.% AAS increased the compressive strength by 23.67% after 60 d sulfate exposure, which was higher than those after 14 and 28 d exposure. This further proved that the hydration reaction of AAS still proceeded at a later age, resulting in a more compact structure. On the contrary, the compressive strength of CSS decreased with increasing exposure times, suggesting more serious damage inside the soil matrix.

### 3.5. EIS

The Nyquist diagram of AASS and CSS at different ages are shown in Figure 9. The ZSimpWin software (version 3.60) was used to fit the Nyquist diagram and the equivalent circuit model is shown in Figure 10. Where R_s_ is the electrolyte resistance in pore solution, R_ct_ is the resistance of charge transfer reaction and C_d_ is the electrode/electrolyte solution interface capacitance and the capacitance of C-S-H gel. In general, only the hydration electrons in the C-S-H gel can carry out a charge transfer reaction. In the paper, because the electrode/electrolyte solution interface capacitance is small, it can be ignored. In addition, the resistance value of R_ct_ is higher than the capacitance value of C_d_, so the parallel resistance of R_ct_ and C_d_ in an equivalent circuit mainly depends on C_d_.

It can be seen that the impedance of the AASS shows a trend of increasing and then decreasing with the increase in the doping amount, and reaches the largest at 15%. This is primarily because the impedance of the AASS is determined by the resistance R_s_ of the pore solution and the capacitive resistance of the Skalny–Young capacitance in the C-S-H gel. The resistance R_s_ of the pore solution of AASS decreases with the increase in Na^+^ content, and the capacitive reactance of the Skalny–Young capacitance in the C-S-H gel increases with the increase in C-S-H gel content. The increase in AAS admixture results in an increase in the contents of Na^+^ and hydration products (C-S-H and C-A-H gels) within the cured soil system. Therefore, the resistance R_s_ of the pore solution decreases and the capacitive resistance of the Skalny–Young capacitance increases. The interaction of the two causes the impedance of AASS to show a trend of increasing and then decreasing with the increase in admixture.

In contrast, the impedance of CSS increases as the amount of admixture increases. This is mainly owing to the fact that, when the cement admixture increases, the hydration reaction intensifies and the hydration products increase, causing a more obvious reduction in total porosity and a greater contribution to the impedance. The Nyquist diagram of CSS is always shifted to the right compared with that of AASS, and the impedance is relatively larger, proving that, during the preliminary hydration process, the products formed by cement hydration are more easily filled with soil porosity compared with those of AAS, resulting in relatively larger impedance.

### 3.6. MIP

Figure 11 depicts the pore size distribution and cumulative porosity curves of AASS20 and CSS20. It can be seen from Figure 11a that the peak pore diameter existed at about 10^2^ nm for AASS20 at 28 d, while the peak pore diameter slightly shifted towards a larger diameter at the age of 90 d. The overall mercury-intruded volume of AASS20-90d was substantially lower than that of AASS20-28d, indicating that the hydration reaction of AAS still continued at 90 d, contributing to a more compact soil matrix. Figure 11b shows that the cumulative porosity of AASS20 decreased with increasing curing ages as a result of the latent hydraulic activity of AAS. The decreased porosity was responsible for the lower water and chloride permeability of AASS samples at a later age. The cumulative porosity of CSS at 90 d was 32.57%, which was much higher than the cumulative porosity of AASS at 90 days, which was 26.10%, further confirming that AAS was beneficial for later-age hydration.

The paste derived from the hydration reaction of the binder can fill the voids between the solid particles in the soil matrix [42]. The volume fraction of the voids before hydration is defined as the original porosity and can be estimated using the following equation:(3)$$P_{0}=\frac{\frac{W/B}{\rho_{w}}}{\frac{f_{p}}{\rho_{p}}+\frac{f_{b}}{\rho_{b}}+\frac{w/b}{\rho_{w}}}$$
where *P*_0_ is the original porosity (%), *W/B* is the water-to-binder ratio, *f_p_* and *f_b_* denote the mass percentage of cement and slag in the binder system, respectively, and *ρ_p_*, *ρ_b_* and *ρ_w_*, respectively, represent the densities of cement, slag and water.

Table 6 shows the original porosity before and after hydration of AASS20 and CSS20. It is clear that, with increasing curing ages, the measured porosity decreased, owing to continuous hydration. Compared with the original porosity, the 90 d porosity of AASS20 decreased by 55.65%, while that of CSS decreased by 46.07%. Due to the secondary hydration reaction of AAS, a large amount of C-S-H gels formed in the soil matrix, which were interwoven with other hydration products to form a denser structure.

Besides the total porosity, the pore volume fraction also plays an important role in governing the transport properties of a porous medium. Small pores can restrain the chloride transport by providing one-dimensional pathways with high tortuosity, while large pores tend to coalesce to provide convenient transport channels with high connectivity. The pores are divided into three types according to their diameters: harmless pores (<20 nm), less harmful pores (20 nm < d < 50 nm) and harmful pores (>50 nm). As shown in Table 7, compared with CSS20-90d, AASS20-90d exhibited a lower fraction of harmful pores, accounting for its better mechanical properties and durability.

### 3.7. XRD

Figure 12 shows the XRD patterns of AASS10 and AASS20 at different curing ages. Quartz, mullite and hematite phases were identified due to the presence of unreacted slag [43]. A wide peak was observed near 28°, which was thought to be calcium hydroxide and poorly crystallized aqueous C-S-H gel [38]. In AASS, alkali activator can destroy the network structure of the glass phase in slag. Thus, the network-forming bonds (Si-O and Al-O bonds) are more prone to break. The combination of AAS with soil particles leads to the formation of high-stiffness C-S-H gels, which is beneficial for strength improvement in soil. Furthermore, the formation of the gel phase also strengthens the solid matrix of the AASS system, making the structure more compact.

### 3.8. Significance of AAS in Cleaner Production of Soil

The use of stabilized soil in construction engineering represents a viable way to reduce engineering costs and environmental pollution. However, traditional stabilization agents, such as cement and lime, are energy-intensive. Blast-furnace slag is a by-product derived from the production of smelting pig iron. If not utilized in an environmentally friendly manner, the discharged slag will not only occupy a large amount of land for disposal, but also cause serious ground water pollution. The ensuing environmental and economic issues will bring serious threats to the eco-system and to the sustainable development of our society. This study indicates that, under the activation of alkali, waste slag is a preferable stabilization agent for soil. The utilization of waste slag and soil proves to be a feasible way to minimize the carbon footprint of the cement industry. The use of waste in this study accords with the concept of sustainable development.

Besides the environmental issue, this study also demonstrates that, compared with cement, AAS is more effective in enhancing the mechanical performance of soil, especially at later ages. A noticeable improvement in sulfate and chloride resistance, reduction in water permeability and refinement in pore structure is also achieved, all pointing to better durability. Apparently, the application of AAS in soil holds great promise for cleaner soil production and turning industrial wastes into valuable resources.

In this work, only the short-term performance of alkali-active slag-solidified soil was studied, and the test section of the solid engineering was not designed; monitoring to verify the actual use of alkali-active slag-solidified soil performance, and its long-term performance changes, can be investigated in future.

## 4. Conclusions

In this study, AAS was used as a stabilization agent for soil, and its effects on compressive strength, water permeability coefficient, chloride migration coefficient and sulfate resistance were evaluated. The following conclusions can be drawn:

1.The later-age compressive strength of AASS is higher than that of CSS at an AAS dosage of above 15 wt.%, which is attributed to the latent hydraulic of slag. A low dosage of AAS is insufficient in inducing the strength enhancing effect.2.Compared with CSS, AAS leads to a lower chloride and water permeability at later-age at high dosages. The sulfate resistance of AASS is much higher than that of CSS, regardless of AAS dosages.3.The impedance of AASS increases with the increase in age; with the increase in AAS content, the impedance of AASS increases first and then decreases.4.MIP analysis demonstrates that the addition of AAS leads to a 6.47% reduction in total porosity and refinement of pore structure by lowering the fraction of harmful pores, which is translated into deceased chloride and water permeability.5.Compared with cement, AAS proves to be a more preferable stabilization agent. The combined use of soil and AAS is a viable method of waste utilization and carbon emission reduction, providing new insights into cleaner production in infrastructure construction.

## Figures and Tables

**Figure 1 materials-16-04803-f001:**
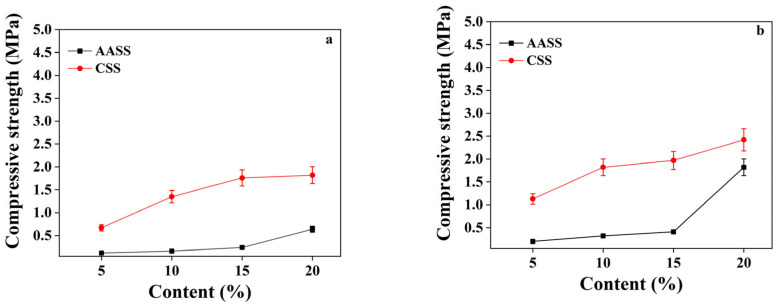
Compressive strength of AASS and CSS at: (**a**) 3 d, (**b**) 14 d, (**c**) 28 d and (**d**) 90 d.

**Figure 2 materials-16-04803-f002:**
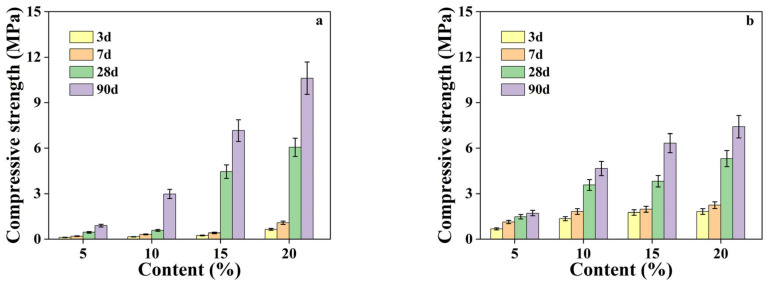
Compressive strength of samples (**a**) AASS and (**b**) CSS at different curing ages.

**Figure 3 materials-16-04803-f003:**
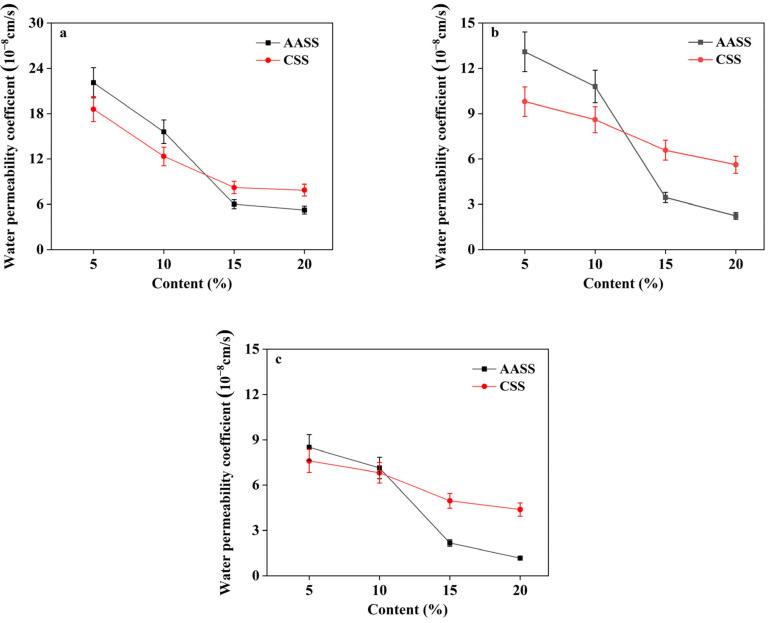
Water permeability coefficients of AASS and CSS at different curing ages: (**a**) 28 d; (**b**) 60 d; (**c**) 90 d.

**Figure 4 materials-16-04803-f004:**
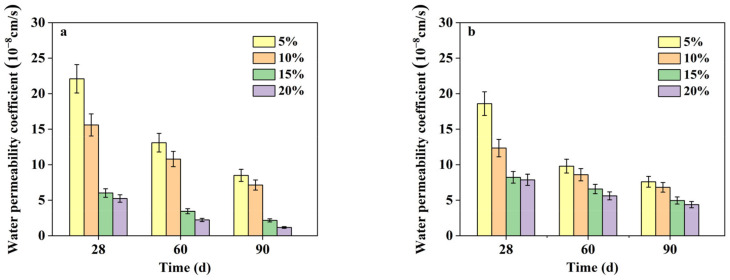
Water permeability coefficients of samples (**a**) AASS and (**b**) CSS at different curing ages.

**Figure 5 materials-16-04803-f005:**
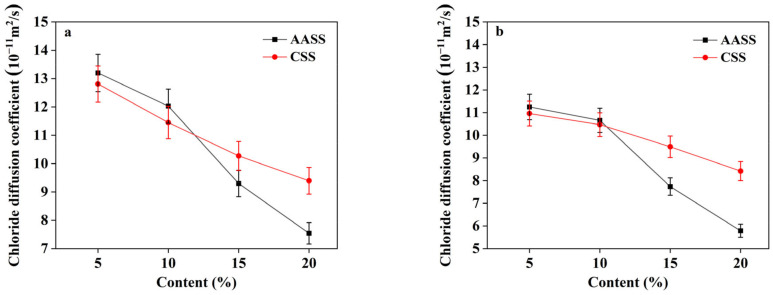
Chloride migration coefficients of AASS and CSS at different curing ages: (**a**) 28 d; (**b**) 60 d; (**c**) 90 d.

**Figure 6 materials-16-04803-f006:**
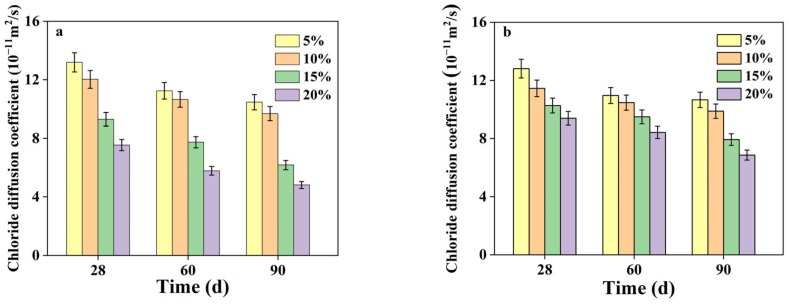
Chloride migration coefficients of samples (**a**) AASS and (**b**) CSS at different curing ages.

**Figure 7 materials-16-04803-f007:**
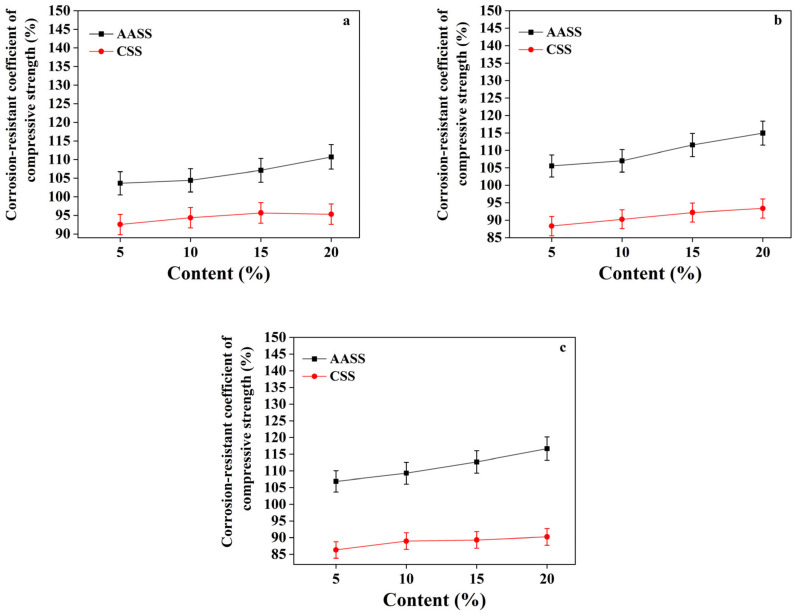
Sulfate resistance coefficients of AASS and CSS at different curing ages: (**a**) 14 d; (**b**) 28 d; (**c**) 60 d.

**Figure 8 materials-16-04803-f008:**
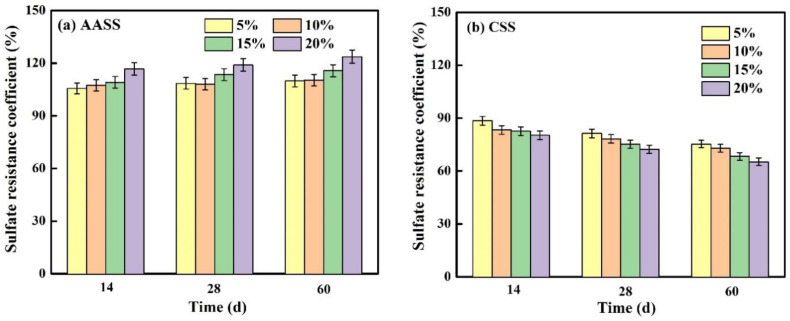
Sulfate resistance coefficients of samples (**a**) AASS and (**b**) CSS at different exposure times.

**Figure 9 materials-16-04803-f009:**
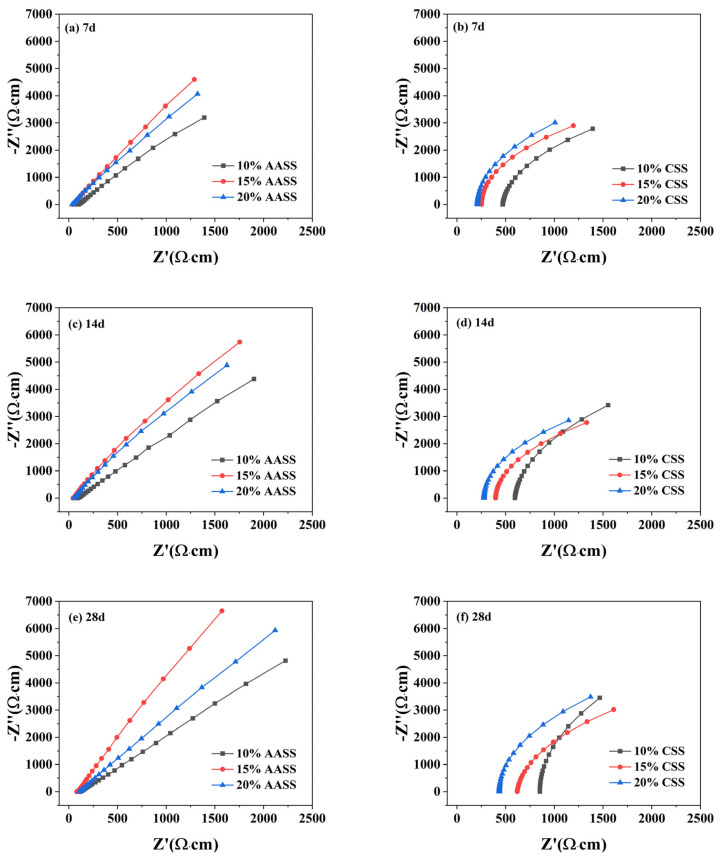
The Nyquist diagram of AASS and CSS at different curing ages: (**a**,**b**) 7 d; (**c**,**d**) 14 d; (**e**,**f**) 28 d.

**Figure 10 materials-16-04803-f010:**
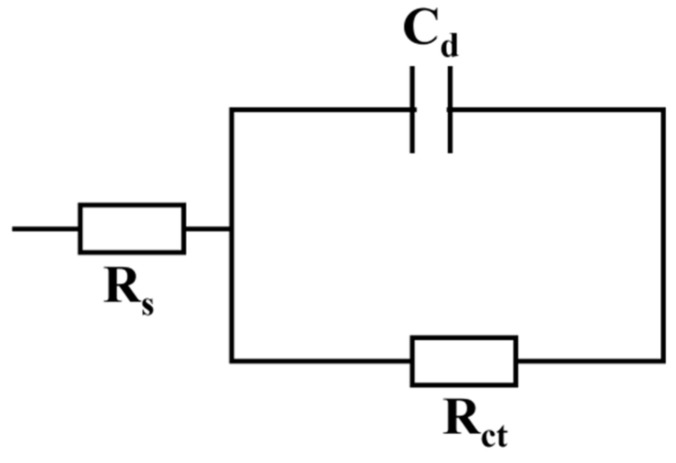
The equivalent circuit model of Nyquist diagram.

**Figure 11 materials-16-04803-f011:**
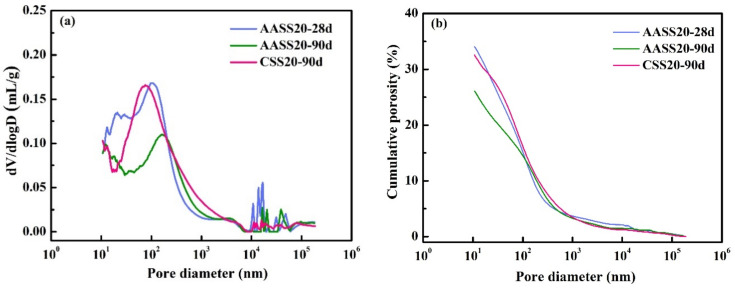
Pore structure of AASS20 and CSS20: (**a**) pore size distributions; (**b**) cumulative porosity.

**Figure 12 materials-16-04803-f012:**
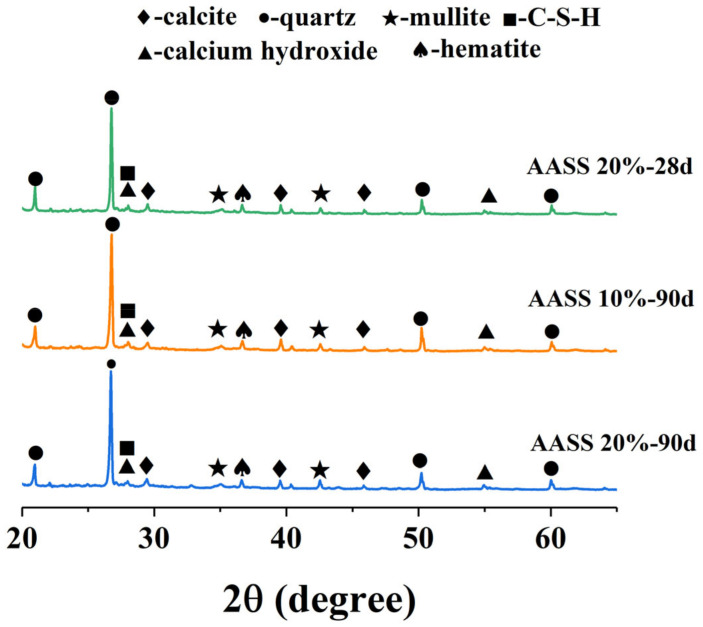
XRD patterns of samples AASS10 and AASS20.

**Table 1 materials-16-04803-t001:** Basic properties of cement.

Compressive Strength (MPa)		Flexural Strength (MPa)		Initial Setting Time (min)	Final Setting Time (min)	Water Requirement of Normal Consistency (%)
3 d	28 d	3 d	28 d			
16.3	35.6	3.2	6.7	230	310	26.8

**Table 2 materials-16-04803-t002:** Basic properties of soil.

Water Content (%)	Void Ratio	Natural Gravity (kN·m^−3^)	Specific Gravity	Saturation (%)	Liquid Limit (%)	Plastic Limit (%)	Plastic Index	Liquid Index
20.1	0.611	18,500	2.72	80	26.5	14.7	12.6	0.15

**Table 3 materials-16-04803-t003:** Chemical composition of slag (%).

SiO_2_	Fe_2_O_3_	Al_2_O_3_	TiO_2_	CaO	MgO	Residue
33.27	2.47	12.17	0.48	39.98	1.38	3.58

**Table 4 materials-16-04803-t004:** Chemical composition of sodium silicate (%).

SiO_2_	Na_2_O	H_2_O
26.4	8.7	61.7

**Table 5 materials-16-04803-t005:** Mix proportions of alkali-activated slag (AASS) and cement-stabilized soil (CSS).

Sample Codes	AAS (wt.%)	Soil (wt.%)	W/B	Sample Codes	Cement (wt.%)	Soil (wt.%)	W/B
AASS5	5	100	0.5	CSS5	5	100	0.5
AASS10	10	100	0.5	CSS10	10	100	0.5
AASS15	15	100	0.5	CSS15	15	100	0.5
AASS20	20	100	0.5	CSS20	20	100	0.5

**Table 6 materials-16-04803-t006:** Original porosity before hydration and porosity after hydration of AASS20 and CSS20.

Sample Codes	Original Porosity (%)	Porosity (%)
AASS20-28d	58.85	34.03
AASS20-90d	58.85	26.10
CSS20-90d	60.39	32.57

**Table 7 materials-16-04803-t007:** Volume fraction of different pores in samples AASS20 and CSS20.

Sample Codes	<20 nm	20 nm~50 nm	>50 nm
AASS20-28d	14.15	16.69	69.16
AASS20-90d	13.36	23.05	63.59
CSS20-90d	10.08	18.95	70.97

## Data Availability

The data used to support the findings of this study are available from the corresponding author upon request.

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
