# Peer review of "Use of Alkali-Activated Slag as an Environment-Friendly Agent for High-Performance Stabilized Soil"

_materials, 2023, doi:10.3390/ma16134803_

Round 1

Reviewer 1 Report

My suggestions is to try to find other paper similar, to discuss abour the % of the alkali-activated slag.

I think is important put the Chemical composition and DRX of  the soil!

Reviewer 2 Report

The paper is about use of alkali-activated slag as an environment-friendly agent for high performance stabilized soil.

The paper is of good quality. Some suggestions, the authors should consider improving the quality of manuscript:

 Abstract:

Follow the basic pattern of template of abstract formation….as:

We strongly encourage authors to use the following style of structured abstracts, but without headings: (1) Background: Place the question addressed in a broad context and highlight the purpose of the study; (2) Methods: Describe briefly the main methods or treatments applied; (3) Results: Summarize the article's main findings; and (4) Conclusions: Indicate the main conclusions or interpretations. (5) Add something about the benefits results of the research.  ……also give quantitative results in the abstract. 

 [Please revise-according to point 4 and 5 Conclusion and benefits]

 Introduction:

 -Review/References should be more than 60. A lot of research is available on this topic [Please add-latest references 2020-2023]

-Can you please establish the research gap [Please add]

Please Add Tables along with citation in the manuscript

(a) Previously used Variables (b) New Variables to be studied (c) Previously asked Questions (d) Previously used techniques for similar type of research…(V. imp)

-Establish your research questions in this section.

-So, establish a research gap and connect your research methodology, data analysis, results with that research gap and produce a discussion on future directions.

 Experimental program

-A comprehensive research framework missing- to follow the research is steps are missing. Add framework-flowchart and write this section in stepwise pattern.

. This portion should also be written in step wise pattern so that readers can understand the procedure for implementation purpose. [Please add]

Discussion.

This section must contain implications for research, practice and/or Field: Does the paper identify clearly any implications for research, practice and/or society? Does the paper bridge the gap between theory and practice? How can the research be used in practice (economic and commercial impact), to influence technical policy, in research (contributing to the body of knowledge)? Add something for field professionals. [Please add and revise section 3.8]

 Limitations of the study:

Please add as heading about the limitations of the study.

Regards

Reviewer 3 Report

Journal:Materials (ISSN 1996-1944)

Manuscript ID : materials-2441843

Title: Use of alkali-activated slag as an environment-friendly agent for high performance stabilized soil

Abstract

Discharged slag not only occupies a large amount of land for disposal, but also causes serious environmental pollution. Use of alkali-activated slag (AAS) instead of cement as a soil-stabilization agent is beneficial for industrial waste disposal and energy conservation, which complies with the concept of green and low-carbon sustainable development in the construction industry. In this study, the compressive strength, water permeability coefficient, chloride migration coefficient and sulfate resistance of alkali-activated slag stabilized soil (AASS) were evaluated, and compared with those of cement stabilized soil (CSS). The hydrated crystalline phases and microscopic pore structures were analyzed by X-ray diffraction, electrochemical impedance spectroscopy (EIS) and mercury intrusion porosimetry (MIP) tests, respectively. The results indicate that compared with CSS, AASS exhibits a higher compressive strength, lower water permeability, chloride migration coefficient and better resistance to sulfate attack at a later age due to the latent hydraulic of AAS. The stabilization effect becomes more remarkable with increasing the AAS dosage with the optimum dosage higher than 10 wt.%. MIP analysis demonstrates that adding AAS leads to pore refinement and a decrease in porosity. The combined use of soil and AAS proves to be a viable and sustainable way for waste utilization and carbon emission reduction in the construction industry.

All components present;Well‐written, concise, clear; Subject matter is original

Introduction:

Minor rewriting is still needed. The hypothesis is clearly presented and is supported by text

Remark:

To alleviate the environmental burden and realize the sustainable development of society, the utilization of resource-saving and environment-friendly engineering materials is required.

If not utilized in an environmentally friendly manner, the discharged slag will not only occupy a large amount of land for disposal but also cause serious groundwater pollution. The ensuing environmental and economic issues will bring serious threats to the ecosystem and sustainable development of our society.

I wonder if an alkali-activated slag-stabilized soil (AASS) will not result in a leaching problem. You should discuss this problem in your paper.

At this moment you can not state that this is an environmentally friendly solution, please comment.

What about leaching tests, ecotoxicity and LCA?

Wan-lu Zhang, Lun-yang Zhao, Zai-jian Yuan, Ding-qiang Li, Liam Morrison, Assessment of the long-term leaching characteristics of cement-slag stabilized/solidified contaminated sediment, Chemosphere, Volume 267, 2021, 128926, ISSN 0045-6535, https://doi.org/10.1016/j.chemosphere.2020.128926. (https://www.sciencedirect.com/science/article/pii/S0045653520331234)

Experimental Program

Adequately written, although writing could be polished; minor typos/grammar/punctuation errors.

The description of procedures needs clarification (clearly remediable), would be difficult for others to reproduce the study by reading the article.  Add information about sample preparation, mixing, curing…

Remark:

Statistical significance of properties is missing (table: 1,2, 3, 4,5)

General remarks add information about the accuracy of your measuring devices.

Results and discussion:

The results support the hypothesis/research question/items mentioned, but rewriting can improve the clarity.

Remark:

The statistical significance of the results are missing. Need to be fixed.

Try to be more concise in selecting the scales of your graphs.

Figure 1. Compressive strength of AASS and CSS at: (a) 3 d; (b) 14 d; (c) 28 d and (d) 90 d.

Use 5 Mpa as max in figures 1a & 1 b.

Use 15 Mpa as max in figures 1 c & 1 d.

Figure 2. Compressive strength of samples (a) AASS and (b) CSS at different curing ages. Use 15 Mpa as max.

Figure 3. Water permeability coefficients of AASS and CSS at different curing ages:.(a) 28 d; (b) 60 d; (c) 90 d. Use 15 and 30 as max.

Figure 5. Chloride migration coefficients of AASS and CSS at different curing ages:.(a) 28 d; (b) 60 d; (

c) 90 d. Use 15 as max.

Figure 7. Sulfate resistance coefficients of AASS and CSS at different curing ages:.(a) 14 d; (b) 28 d; (c) 60 d. Use 150 as max.

Figure 9. The Nyquist diagram of AASS and CSS at different curing ages:.(a-b) 7d; (c-d) 14 d; (e-f) 28 d. Use 7000 and 2500 as max.

Figure 10. The equivalent circuit model of Nyquist diagram. Is this figure adding extra information, is it needed.

Figure 12. XRD patterns of samples AASS10 and AASS20. Improve, difficult to read.

3.8. Significance of AAS in cleaner production of soil

If not utilized in an environmentally friendly manner, the discharged slag will not only occupy a large amount of land for disposal, but also cause serious groundwater pollution.

Are you sure that the proposed method is environmentally friendly? Can you comment on leaching and ecotoxicity?

What about an LCA?

4. Conclusions

Statements and conclusions are presented but need minor revisions to correlate with data and link with goals

Study implications and/or limitations are not presented, Please add them (Leaching/EcoTox)

Minor rewriting is still needed.

minor typos/grammar/punctuation errors

Reviewer 4 Report

In the modern world, a very strong focus is being placed on the problem of reducing the amount of CO2 into the atmosphere.  On the one hand, the materials used in construction should be environmentally friendly, while on the other hand, they should meet the requirements set by standards. Geopolymers seem to fit perfectly into these trends. They are materials characterised by high compressive and flexural strength, very high acid resistance and resistance to chlorides and sulphates, resistance to weathering, including very high frost resistance, high thermal resistance, faster onset of setting compared to concretes, dimensional stability (no or little shrinkage on setting), synthesis of geopolymers consumes 2-3 times less energy than Portland cement and releases 4-8 times less CO2.

The authors have given due consideration to the subject matter presented. However, due to the fact that the authors used slag in their composite preparation, in my opinion it should be talking about geopolymer composites here and not alkali-activated slag. Of course, it is related, while the specific three-dimensional struture of the geopolymers influences all those physical and chemical parameters mentioned by the authors. I would advise you to reflect on the nomenclature.

Topic of the publication - to be rethought by the authors. The language used in the article is correct references sufficient

The article needs minor editorial correction:

1. centring of figures,

2. value and unit should be on one line.

To conclude, the topic area related to the application of new materials with improved properties as soil improvement is very interesting and I hope that it will be implemented in practice. The article has the hallmarks of a scientific publication and I recommend it for publication in the journal Materials, but once again I would ask the authors to think about the terminology used.
